# Reward systems for cohort data sharing: An interview study with funding agencies

**Thijs Devriendt**[1]*, **Mahsa Shabani**[2], **Pascal Borry**[1]

**1** Department of Public Health and Primary Care, Centre for Biomedical Ethics and Law, KU Leuven, Leuven, Belgium, **2** Faculty of Law and Criminology, METAMEDICA, UGent, Ghent, Belgium

* thijs.devriendt@kuleuven.be

## Abstract

Data infrastructures are being constructed to facilitate cohort data sharing. These infrastructures are anticipated to increase the rate of data sharing. However, the lack of data sharing has also been framed as being the consequence of the lack of reputational or financial incentives for sharing. Some initiatives try to confer value onto data sharing by making researchers' individual contributions to research visible (i.e., contributorship) or by quantifying the degree to which research data has been shared (e.g., data indicators). So far, the role of downstream evaluation and funding distribution systems for reputational incentives remains underexplored. This interview study documents the perspectives of members of funding agencies on, amongst other elements, incentives for data sharing. Funding agencies are adopting narrative CVs to encourage evaluation of diverse research outputs and display diversity in researchers' profiles. This was argued to diminish the focus on quantitative indicators of scientific productivity. Indicators related to open science dimensions may be reintroduced if they are fully developed. Shifts towards contributorship models for research outputs are seen as complementary to narrative review.

**Data Availability Statement:** All anonymized transcript data, except for one, are available in the Social Sciences and Digital Humanities Archive (SODHA): doi.org/10.34934/DVN/JCMXTY.

## Introduction

Data sharing platforms are being constructed to facilitate the sharing of data from cohort studies in biomedical research. These platforms combine various technical components such as data catalogues, data access management systems and virtual research environments. These components are intended to contribute to the findability, accessibility, interoperability, and reusability (FAIR) of cohort studies [1, 2]. More specifically, they contribute to FAIRness of cohort studies in practical terms rather than FAIRness in terms of minimal machine-readable metadata standards. Cohort studies consist out of privacy-sensitive health data that are subject to local and international data protection regulations, such as the General Data Protection Regulation (GDPR) of the European Union [3]. They can be made available under controlled access models, where data applicants need to submit and motivate their data access requests. Presently, research teams that have cohort studies often organize their own data access request systems. These systems may be formal application interfaces on websites or simply contacting via personal email. This complicates reusing data from multiple cohort studies for research

**Funding:** This publication is part of a project that has received funding from the European Union's Horizon 2020 research and innovation programme under grant agreement No 825903 (https://ec.europa.eu/programmes/horizon2020/en). The funders had no role in study design, data collection and analysis, decision to publish, or preparation of the manuscript.

**Competing interests:** The authors have declared that no competing interests exist.

projects as requests are sent out to multiple unique application systems [4]. In this sense, data sharing platforms can be seen as overarching infrastructures that connect and facilitate access to multiple cohort studies. Platforms exist in multiple forms and sizes. They may operate on national or international levels. Platforms may host generalist or disease-specific collections of studies. They may have central or decentralized data governance arrangements. Some examples of platforms include the European Medical Information Framework (EMIF) platform, Dementias Platform UK (DPUK), the Swiss Personalized Health Network (SPHN), Health-RI and euCanSHare [5, 6].

These platforms are often framed as technical solutions that speed up data sharing. Nevertheless, there are many publications and policy documents that emphasize that stimulating data sharing requires creating incentives for researchers to share. For instance, Sim *et al.* describe that "*researcher reluctance to share is a rational response to existing incentive systems that measure and reward individual achievement partly on the basis of the accumulation and use of closely held data sets*" [7]. This reasoning is based on the idea that the behavior of researchers is, at least partly, motivated by their own self-interest. External factors, such as promotion schemes at research institutions and conditions for grant acquisition, determine what is in researchers' self-interest. The term "incentive" has its roots in economic thinking and can be defined as "a thing that motivates an actor to behave in a particular way". Incentives are thought to be important for altering these external conditions that motivate behavior. These incentives can take various forms. For instance, incentives can be *financial* or *reputational*. Financial incentives might be compensations for costs incurred when sharing data. Reputational incentives might be things that change how much recognition researchers acquire for certain activities. In reputation economies, like academia, reputational incentives also act as financial incentives. This is because reputation increases the availability of resources, which in turn, may be used to gather more reputation (giving rise to the Matthew-effect). From this perspective, creating incentives is necessary to align researchers' self-interest with the public interest. This idea is incorporated in many open science policy documents that call for reforms to reward systems, such as the San Francisco Declaration on Research Assessment (DORA) [8–10].

Several evolutions are taking place that could be seen as creating reputational incentives for data sharing. First, attribution models can be created that better capture the broad range of contributions to scientific articles and other outputs. These include the Contributor Role Taxonomy (CRediT), Contributor Role Ontology (CRO) and Data Authorship [11–15]. These designations allow attributing more diverse contributions to individuals than under the ICMJE authorship model. Second, indicators could be developed that relate to research processes or broad scientific productivity. In terms of data sharing, these may reflect data reuse (e.g., data citations), FAIRness (e.g., FAIR Maturity Assessment) or openness [16, 17]. We have previously illustrated how data access request records of cohort studies could be used to understand data sharing practices of cohort holders and help in assessing the value of platforms and cohort studies themselves [18]. These indicators should be linked to both projects (e.g., cohort studies) and, ideally, those who are responsible for them. The use of both attribution models and indicators in evaluation systems could incentivize data sharing by conferring more value onto data production and sharing. Reputational incentives may need to be complemented by financial incentives, such as appropriate cost-recovery models for data sharing [19, 20].

Even if reputational incentives are fully developed, it is unclear how these mechanisms would be interpreted and used within grant and tenure track evaluation. Many different levels of discussion are taking place of how research evaluation systems themselves should be designed. These discussions touch upon the relation between quantitative indicators and

qualitative assessment, shifts in the "level of evaluation" in science, timelines and labor-intensiveness of grant review, and the use of process and output indicators [8, 10, 21–23]. The ideal scenario is that attribution models and indicators are tailored in advance to "fit" within downstream evaluation and funding distribution systems.

We need to assess whether there is congruence or divergence between proposals made (e.g., specific attribution models, indicators) in scientific literature and the policy orientations of funding agencies. For this reason, it is important to explore the perspectives of funding agencies on intended reforms of academic evaluation systems. In this way, the viability of these proposals for implementation can be gauged.

The goals of this study were therefore to document views of funding agencies on (1) potential alterations to recognition systems in academia; (2) incentives to enhance the sharing of cohort studies; (3) data sharing policies in terms of the governance of cohort studies; (4) other potential interactions between science policy and data sharing platforms for cohorts. This article focuses on creating incentives for data sharing through altering attribution and evaluation systems in academia. The results obtained from this interview study on policy measures that somewhat restrict decision-making options for researchers (e.g., mandates) are reported elsewhere [24]. These policy measures are complementary to the design of reputational and financial incentives within the overall science policy framework.

## Methods

### Recruitment

Semi-structured interviews were performed using an interview guide. Funding agencies were selected using a purposive sampling strategy. Funding agencies were selected to acquire sufficient representation of (a) national and international agencies; (b) public and philanthropic agencies; and (c) agencies in continental European and the anglophone world. Suitable funding agencies were selected from a list of health research funding organizations according to their annual expenditure on health research (https://www.healthresearchfunders.org/health-research-funding-organizations/) and via targeted internet searches. For selected organizations, general email addresses were retrieved via their websites. When no suitable general email addresses were easily retrievable from websites, contact persons were selected via lists of employees according to their responsibilities. Alternatively, names of contact persons were extracted from the lists of *Science Europe* Working Groups (e.g., Data Sharing and Supporting Infrastructure Group). Lastly, some contact persons were identified via personal connections with members of these agencies. Inclusion criteria listed in the contact email included speaking English; having some knowledge on the development of novel incentives for data production or sharing (e.g., citations, data usage metrics. . .); having knowledge on data sharing policies (e.g., mandates for sharing, data access committees, data management plans. . .); and being familiar with data sharing procedures for medical research data (e.g., controlled-access models, privacy-sensitivity. . .). Due to the difficulty of individual members of agencies meeting all these criteria, multiple members were allowed to participate per agency. Prospective participants that declared to possess general knowledge on data sharing procedures, but not specifically on medical research data, were allowed to participate. All participants were invited to participate in semi-structured interviews, which used an interview guide, but where participants could lead the conversations. The sample of funding agencies was not intended to be statistically representative or exhaustive.

In total, members of 17 funding agencies were interviewed. All participants signed a written informed consent. The interview with one member was disqualified due to being of very short length, interrupted and insufficient topics relevant to the objectives being discussed. Funding

**Table 1. Funding agencies of interviewees enrolled, classified according to the type of organization, and the range and scope of funding activities.**

| Organization | Geographical range of activities | Scope of funding activities | Type of organization |
|---|---|---|---|
| Interview 1 | National/regional | Generalist | Public agency |
| Interview 2 | National/regional | Generalist | Public agency |
| Interview 3 | International | Specialist | Foundation |
| Interview 4 | National/regional | Generalist | Public agency |
| Interview 5 | International | Generalist | Public agency |
| Interview 6 | National/regional | Specialist | Public agency |
| Interview 7 | National/regional | Specialist | Public agency |
| Interview 8 | International | Specialist | Foundation |
| Interview 9 | International | Specialist | Public agency |
| Interview 10 | National/regional | Generalist | Public agency |
| Interview 11 | International | Specialist | Foundation |
| Interview 12 | National/regional | Generalist | Public agency |
| Interview 13 | National/regional | Specialist | Public agency |
| Interview 14 | National/regional | Generalist | Public agency |
| Interview 15 | International | Generalist | Public agency |
| Interview 16 | National/regional | Generalist | Public agency |

agencies were categorized according to the type of organization (public agency [13/16] vs. foundation [3/16]), scope of funding activities (generalist [9/16] vs. specialist [7/16]) and their geographical range (national [10/16] vs. international [6/16]). Characteristics per funding agency can be found in Table 1. The distinction between generalist vs. specialist was made on the basis of the area of funded research activities. No participants were known by the interviewer on beforehand.

### Procedure

Interviews were conducted via Microsoft Meetings or Zoom. All interviews were audio-recorded and transcribed verbatim. Interview transcripts were analyzed through inductive content analysis, in which codes emerge from data rather than being predetermined [25]. NVivo 12 software by QSR International was used for the coding procedure. Iterative coding steps were performed by the first author, with repeated adjustment of coding categories, recoding of text fragments, and revisiting interview transcripts. In this way, codes did not emerge exclusively around topics listed in the semi-structured interview guide. Initial codes were coarse rather than fine-grained due to strong context-specificity of statements. Sequential coding steps involved making most coding categories more fine-grained while some were made broader and merged (due to an inability to fully uncouple them from considerations in other categories).

This article reports on the results of this study related to attribution and evaluation systems for data sharing. The results related to data sharing policies and funding models for data infrastructures and data sharing practices will be covered elsewhere. The study was approved by the Social and Societal Ethics Committee (SMEC) at KU Leuven (G-2021-3823). The full methodological details can be found in the Supplementary Data of the publication made on science policy measures based on this interview study [24].

### Results

Codes that emerged from the interview transcripts were clustered around various themes (see S1 Fig). With respect to recognition systems, the following themes emerged: (a) evaluation

systems for data sharing; (b) alternative attribution systems for data sharing; (c) role of the research community and international alignment of policies.

## a. Evaluation systems for data sharing

Interviewees considered that better mechanisms for rewarding data sharing and management need to be developed. This was often framed as part of a broader reformation of career progression and grant assessment in academia that aims to foster open science activities. Nearly all interviewees explained that their funding agencies had signed the San Francisco Declaration on Research Assessment, which calls for research to be assessed on its own merits rather than by using journal-based metrics.

Many interviewees described the introduction or the intention to introduce narrative CVs by their funding agency. Narrative CVs allow researchers to describe their contributions to science, including their activities in terms of data management, infrastructure creation and software development, in narrative format. Interviewees described this evolution as an attempt at expanding the types of credit-worthy contributions to science, reemphasizing narrative and context in scientific evaluation, and reducing the weight of (proxy) quantitative indicators of scientific productivity.

*We are currently changing our CVs to [put] less focus on publication lists, and more focus on whatever the scientists think are the most important achievements and relevant to the project. Therefore, [we] also focus on alternative outputs like datasets and so forth. [. . .] For now, the tendency is rather to not include any metrics at all. [Our agency] has the goal to not focus on metrics but to look [at] the whole profile. But if certain metrics in the future pop up which might be really valuable and indicate the open science degree of research, [they] might be considered.*

*(Interviewee 1 –public agency/generalist/national)*

Several interviewees stated that they anticipate deeply-entrenched cultural beliefs, such as the conception of publications as the sole output of scientific research, to complicate expanding reporting of research outputs.

*We have for many years said: "Tell us about all of your outputs" but found that the only thing we get back is a list of publications. [. . .] So, there is no longer [going to be] any publication list within the grant application process and it [will be:] Tell us about your contributions to knowledge. . . Then there is another question: "Tell us about your contributions to the research environment." And for the most senior ones, there [will be]: "Tell us about your leadership and mentoring". So, we are really trying to throw open the door to enable people to showcase their contributions. [. . .]. I think there is an issue within academia. There is this mindset that publications are the only thing that count, [and] therefore I need to have publications.*

*(Interviewee 3—foundation/specialist/international)*

The evaluation of narrative CVs was raised to pose cultural and practical barriers for academics. Interviewees raised that committee members may find narratives presented by grant applicants difficult to compare. Furthermore, committee members may fear that narratives introduce more "subjectivity" into evaluation processes. Lastly, the writing and peer-review of grant applications may become labor and resource-intensive as there is little experience with these processes. To overcome these barriers, the role of educational efforts on the side of both grant applicants and evaluators was emphasized by several interviewees.

*I hear that committee members find it difficult to evaluate and compare people because of the narrative. [. . .] It is a narrative and not metric. They are afraid that it becomes more subjective. [. . .] I am not directly involved in this but these are the things I am hearing. Some people are not happy that certain metrics cannot be used anymore and committee members having trouble how to evaluate. Applicants struggling with the narrative [. . .]. It is about marketing and selling themselves.*

*(Interviewee 10 –public agency/generalist/national)*

Various interviewees elaborated on the balance between the use of quantitative indicators and qualitative assessment (e.g., by peers). This brought forward various positions. First, it was proposed that indicators can be framed *within* narrative CVs on the condition that they directly relate to research outputs. Second, indicators could be used within scoring systems that are operated in parallel to narrative CVs. In this case, standardized scoring schemes take into account novel indicators in addition to other established indicators. Third, indicators on research processes related to open science practices could be developed and implemented in the long-term. Lastly, indicators are more relevant when evaluating institutions and regions rather than individual researchers or research teams. In this view, the use of traditional indicators may be more relevant to evaluate higher-level entities (e.g., institutions, regions. . .) rather than individual researchers. Nevertheless, interviewees often raised that it remains unresolved how any alternative indicators will exactly be implemented in the future. Notably, several interviewees raised that, for their funding agency, the mentioning or improper use of journal-based metrics will be prohibited.

*In that narrative CV, it is not that people cannot use metrics. It is just that people have to use object-level metrics. You are not to use journal level metrics, H-index. . . You can cite metrics at the research output level: My paper or dataset has been cited this many times. [We do not allow] metrics that are not good proxies for what you are trying to convey. [. . .] But you always have to put it into context because a metric alone can always be played with in a way. I think in our implementation of DORA, it is really about metrics plus context plus narrative.*

*(Interviewee 10 –public agency/generalist/national)*

Interviewees saw the principal advantages of the use of indicators as being less time-intensive for grant evaluators, easy to rank and that they made methods more objective. The principal disadvantages of the use of existing metrics were argued to be that they cannot be compared across fields, require contextualization, are not understood very well, inconspicuously bias one's judgement, unintentionally restrict behavior of academics, elicit detrimental behavior, and have unclear links to the probability of successfully completing submitted projects in grant evaluation.

One interviewee argued that the link between past actions by grant applicants on data sharing and the likelihood of receiving funding was absent in some cases. For longitudinal cohorts that are considered large investments, grant applicants may already be asked to provide information on past data sharing practices of cohort studies. This information would be reviewed and inform the provision of funding to extend the lifespan of cohort studies. In contrast, some interviewees argued that such information would not be requested in grant applications for *novel* data collections. Consequently, whether grant applicants did or did not engage in data sharing does often not impact funding provision.

## b. Alternative attribution systems for data sharing

Two alternative attribution systems were commonly discussed throughout the interviews: the use of the contributor roles taxonomy (CRediT) and data citation. Interviewees considered that CRediT had potential to be further developed and implemented. CRediT was described as complementary to narrative CVs, at least in terms of those contributions related to scientific publications. Furthermore, CRediT was argued by some interviewees to be useful to adequately recognize research support staff, such as engineers or data specialists. Nevertheless, questions were raised on the discipline-specific relation between authorship and contributorship, and on the shortcomings of the model itself (e.g., power differentials in deciding upon contributions). The (quantitative) evidence basis that could support the use of alternative attribution systems was argued to be scarce. Some agencies already accept the submission of contributor statements in their funding application forms. The use of CRediT was also mentioned as being useful in assessing the inclusion of researchers in scientific partnerships between high-income and low-middle income countries (HICs-LMICs). Additionally, CRediT was raised to be potentially useful in future "team-science" approaches, where principal investigators would be required to demonstrate that they possess sufficient expertise to conduct all activities within the proposed research project. Data citation was also generally supported by interviewees, which reported that citations for research data in general repositories could, in principle, be used for evaluation purposes in the future.

> *Even if we would ask [contributorship information] and presented it to reviewers of proposals, they would not know what to do that. What does it mean now [that] the[ir] contribution is this? [. . .] These kinds of things, it just needs time. I think we as funders, we could definitely start discussions around it but it is something that needs to be done by a broader community of funders. We need to train the reviewers and [make them] aware of what it actually means, and how they can then evaluate people or proposals based on these contributions.*
>
> *(Interviewee 4 –public agency/generalist/national)*

## c. Roles of research institutions in changing recognition systems

Interviewees generally described the role of their funding agency in policy making as facilitators of bottom-up initiatives. The responsibilities of research communities to organize themself was emphasized in, amongst others, the context of (a) designing alternative attribution systems; (b) altering evaluation systems at universities; (c) raising awareness on institutional support for data sharing; and (d) encouraging data sharing by local researchers. One funding agency described their role as neither top-down or bottom-up, but rather *"a horizontal approach with collective decision making by research institutions to set norms [. . .] and funding agencies to enforce those norms"* (paraphrased). Others clearly emphasized the role of the research community itself.

> *We follow a bottom-up approach. We give them a certain structure but within that structure, we try to encourage that scientific communities define their own standards. [. . .] I think we can obviously give guidelines to communities who are struggling or for which [open science] is a new concept. The main driver should still be the communities unless in twenty years nothing happens. Then we might have to say something.*
>
> *(Interviewee 1 –public agency/generalist/national)*

Interviewees considered that cultural change is fundamental to overhauling the academic attribution and evaluation system. Some members of funding agencies voiced their frustrations that cultural discrepancies exist between the funding agencies and some research communities.

*A large [number] of funders sign on to DORA and are really trying to change the incentives. Yet I still [hear] in discussions that "I need to publish in Nature or I'm not going to get grant funding". From who at this point? [. . .] All of the major funders, especially in Europe, have signaled that we are trying to shift to the intrinsic merit of the research itself and not the venue in which it is published. There is still a lot of stigma. I think it is because institutions have not caught up with funders yet and they are the ones doing the hiring. But I am always so surprised to see in those conversations [that they say] funders should really change this.*

*(Interviewee 11 –foundation/specialist/international)*

Interviewees mentioned that international alignment by funding agencies, research institutions and journals on the attribution and evaluation systems is essential for those systems to be useful in practice.

## Discussion

In this study, we report the views of members of funding agencies towards potential alterations to recognition systems in academia, focusing on data sharing activities. Our results indicate that narrative CVs are being adopted by funding agencies with the intention of valuing *all* research outputs and recognizing diversity in contributions. Novel indicators may be acceptable for use if they are related to open science processes (e.g., data sharing). Such indicators may be used within narrative CVs or grant evaluation more generally. This may include the use of indicators to facilitate qualitative assessment or in quantitative scoring schemes. Funding agencies are generally open towards alternative attribution systems that can be used alongside narrative CVs, such as CRediT. Changing deeply-rooted practices within academic culture and reaching international alignment on attribution and evaluation systems were both understood to be essential to realize open science practices.

### Narrative CVs

Policy documents emphasize that the relationship between qualitative assessment and quantitative indicators of scientific productivity should be recalibrated [21–23]. As stated in the interviews, one important effort in this direction is the piloting of narrative CVs by various funding agencies. These include the Swiss National Science Foundation, Science Foundation Ireland, Luxemburg National Research Fund and the Dutch Research Council [26–29]. Meadmore *et al.* suggest that the format of narrative CVs should be designed in function of the funding streams and awards [30]. This could mean differentiating sections within CVs between types of funding streams. For cohort studies, this might mean differentiating between grants aimed at expanding data infrastructures, grants aimed at extending the lifespan of longitudinal cohorts, grants aimed at setting up new cohort studies and grants aimed at secondary data analyses.

### Contributorship models

Interviewees stated that they saw contributorship models as useful to contextualize researchers' contributions. Contributorship models could be developed for multiple types of outputs.

CRediT, CRO and Data Authorship were designed for publications on empirical and experimental studies [12, 15]. Contributorship models have also been designed for some types of datasets. Zenodo allows attaching DataCite contributor roles to research objects, such as "Datasets" and "Collections" [31]. The EnviDat Consortium creates its own data-oriented role list called DataCRediT for environmental data, which is limited to collection, validation, curation, software, publication and supervision [32]. Aside from contributorship models being developed for different types of outputs, they could also further evolve to be tailored to particular disciplines, adding levels of contributions per category or weighting contributions [33–36].

Some interviewees argued that contributorship models are currently not sufficiently understood to be used in grant evaluation. There are relatively few field-specific studies that relate contributorship statements on publications to authorship [37–39]. There are obviously no studies that explore the use of contributorship statements for other outputs that are not developed yet (e.g., data standards, cohort studies). Furthermore, the logic that guides the development of contributor models is often not made explicit [40]. This raises the question what principles should guide the development of contributor models for any outputs that are not publications, particularly if these models overlap (e.g., datasets ↔ publications using data). Hosseini *et al.* propose the *significant threshold test* to identify core roles in contributorship models, which draws on the concepts of indispensability and specificity [40]. They define these dimensions as "so important that research objectives cannot be achieved without them" and "directly associated with the questions and content of research, and constructively affect the reliability, validity and the justification of the reported data, claims, results and conclusions", respectively. The use of these dimensions for identifying roles means that normative criteria are not completely removed from contributorship models. In this case, credit-worthy activities are not defined by guidelines, but are implicit in in taxonomy development (hence the name: *significant threshold test*).

## Indicators of scientific productivity

Interviewees explained that their agencies are generally moving towards reducing the emphasis on quantitative indicators. Nevertheless, indicators may be mentioned within narrative CVs and those indicators related to open science dimensions may be considered in the future. Some groups have given advice on how these indicators should be developed. In July 2019, the European Commission's Expert Group on Indicators for Researcher's Engagement with Open Science published the *Indicator Frameworks for Fostering Open Knowledge and Practices in Science and Scholarship* report [8]. In this report, the Expert Group proposes that indicator frameworks and toolboxes should be developed. Indicator frameworks relate to contextual dimensions of indicators: goals of evaluation or monitoring, research mission of scientific domain, level of assessment, epistemic cultures within fields, modes of knowledge exchange and the research environment. Indicator toolboxes contain sets of indicators that relate to various open science policy goals. In short, both the contextual dimension and technical operationality of indicators need to be clearly outlined. No existing resources currently allow formulating indicators for cohort data sharing with this degree of conceptual or technical detail.

The most prominent initiative to develop indicators is data citation. Digital object identifiers that allow data citation in publications are already attached to data objects in general repositories [41]. Other initiatives to develop indicators include FAIR Maturity Assessment of minimal metadata standards on data located within these repositories, and information extracted from machine-readable data management plans. We previously suggested that the ability of platforms to monitor data sharing practices and decision-making of data access

committees may make them suitable to formulating indicator toolboxes on data sharing for cohort studies [18]. This would require common data access management systems, such as Data Usage Oversight System (DUOS), to be integrated in multiple platforms [42].

The preference of some interviewees for indicators related to open science processes over typical output indicators could stem from the idea that output indicators are more susceptible to gaming behavior that is detrimental to scientific practices overall. For instance, researchers might try to get few indicators as high as possible (e.g., journal impact factors of journals that publications are made in) at the cost of proper research processes or other less prominent indicators. In this sense, the core flaw of output indicators is that they are unduly "targeted" by researchers, which undermines their use as measuring sticks for performance (i.e., *Goodhart's Law*) [43].

## Interplay between narrative CVs, contributorship models and indicators

As indicated by interviewees, contributorship statements and indicators could be used both *within* or *alongside* narrative CVs. For instance, researchers could use indicators within narrative CVs to back up claims about their data- or software-related outputs with process indicators related to open science. For cohort studies, researchers could refer to platform-generated process indicators [18]. In particular, they could refer to minimal metadata standards of cohort studies within (multiple) data catalogues. Furthermore, they could illustrate that data from cohort studies are being sufficiently shared upon request within reasonable timelines, with multiple types of entities (e.g., students, commercial, academic). If data access records are linked to publications, they could also try to illustrate that sufficiently high numbers of data sharing instances for academic purposes lead to publication, also if testing hypotheses ends in negative results. They could demonstrate that data usage conditions, such as consent codes, have been made machine-readable or available otherwise (e.g., in English). They may also show that data access committee membership and procedures have been made transparent through data access management systems in platforms.

Interviewees stated that indicators may also function alongside narrative CVs, such as into scoring systems in research evaluation. In that case, researchers could be forced to disclose information about their data sharing practices. Overall, this would increase comparability between researchers' profiles, which addresses one of the concerns that interviewees mentioned. The use of indicators within or alongside narrative CVs could tackle the problem mentioned by one interviewee that past data sharing activities are currently not strongly linked to funding acquisition.

Like indicators, contributorship models could be used within narrative CVs. Interviewees raised that contributorship models may help credit those in support roles, and that they may be useful for team-science approaches to research evaluation. This could be done in multiple ways. First, researchers could be asked to demonstrate, if possible, what their contributions are with regards to research outputs. Second, they could be asked to provide their contributor profiles, which should be designed to illustrate the balance between contributions to different outputs (rather than absolute numbers of contributions).

## Limitations to use of contributorship models and indicators

As mentioned by interviewees, there are several important drawbacks to using narrative CVs, contributorship models and indicators. A general problem in narrative CVs is the verifiability of claims made by researchers. Researchers may grossly exaggerate their contributions or provide inaccurate information about their responsibilities. Luxemburg National Research Fund's evaluation of narrative CVs reveals that concerns around their use cluster around four themes:

(a) insufficient space to outline achievements; (b) reviewers can still access and use quantitative information; (c) narrative CVs benefit those proficient in writing; and (d) narrative CVs take more time to both write and evaluate [28]. Narrative CVs could disadvantage non-native speakers, minorities and those without institutional support [44].

Contributorship models have inherent drawbacks. As stated earlier, quantitative studies that explore relations between contributorship and authorship models are few in number. Empirical research indicates that researchers may be concerned about various aspects [19, 45]. Attributing outputs may increase the overall bureaucratic load for researchers. Researchers may interpret roles differently. Contributor models may not be uniformly adopted across their relevant research communities. The lack of machine-readable codes and aggregation mechanisms may deter grant evaluators from using this information. Some forms of team-work may be difficult to divide into contributions to roles (e.g., conceptualization). There may be lack of detail in current models in terms of number of categories or different weights per category. The attribution of outputs may risk introducing conflict between team members. Lastly, contributorship models may still be subject to gaming behavior if they are used in research evaluation (e.g., assigning responsibilities for labor that one did not perform).

One limitation to the use of process indicators on data sharing practices in research evaluation is that they need to be correctly "assigned" to researchers and/or research teams. This requires making decisions on whether individual researchers *should* carry the consequences for those actions that are not fully within their control. In practice, individual responsibilities cannot easily be distinguished from shared responsibilities in hierarchical work environments. It is possible that the use of indicators might only be appropriate to evaluate those assuming final responsibility for the management of datasets. In practice, this might mean holding principal investigators of cohorts and/or those with final responsibility data management and infrastructure units responsible. Another clear limitation is that platform-derived indicators on data sharing require data access management systems to be used amongst platforms. No information can be obtained about those cohort studies that are not uploaded in platforms or that are uploaded in platforms without these monitoring systems.

## Conclusion

Funding agencies are adopting narrative CVs that enable appropriate valuing diverse research outputs and diversity in research profiles. Interviewees suggested that (process) indicators of scientific productivity may be used within narrative CVs or grant evaluation if they relate to open science dimensions. Funding agencies interviewed in this study are generally supportive of further development contributorship models, which is seen as congruent with moves towards greater contextualization of academic accomplishments. Contributorship models for outputs other than publications could be considered, such as for cohort studies. Platforms could generate indicators on cohort data sharing via their data management systems. Contributorship statements and indicators could, be used in explicit claims presented by researchers within their narrative CVs or part of scoring systems within grant evaluation. This requires contributorship models for different outputs to be delineated from each other. This would facilitate checking claims that are presented within narrative CVs. There are various limitations to using indicators and contributor statements within research evaluation. The use of indicators requires assigning responsibility to particular persons, which is difficult within hierarchical environments with a far-reaching division of labor. Contributor models still need to be developed for outputs beyond research publications.

## Supporting information

**S1 Fig. Coding themes.**
(TIF)

**S1 File.**
(DOCX)

## Acknowledgments

We would like to thank all participants in this study for making themselves available to respond to our questions and share their views.

## Author Contributions

**Conceptualization:** Thijs Devriendt, Mahsa Shabani, Pascal Borry.

**Data curation:** Thijs Devriendt.

**Formal analysis:** Thijs Devriendt.

**Funding acquisition:** Mahsa Shabani, Pascal Borry.

**Investigation:** Thijs Devriendt.

**Methodology:** Thijs Devriendt, Mahsa Shabani, Pascal Borry.

**Project administration:** Thijs Devriendt, Mahsa Shabani, Pascal Borry.

**Supervision:** Mahsa Shabani, Pascal Borry.

**Writing – original draft:** Thijs Devriendt.

**Writing – review & editing:** Thijs Devriendt, Mahsa Shabani, Pascal Borry.

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
