## [Decision Letter · Decision Letter 0]

12 Dec 2022

PONE-D-22-23763Reward Systems for Cohort Data Sharing: An Interview Study with Funding AgenciesPLOS ONE

Dear Dr. Devriendt,

Thank you for submitting your manuscript to PLOS ONE. After careful consideration, we feel that it has merit but does not fully meet PLOS ONE’s publication criteria as it currently stands. Therefore, we invite you to submit a revised version of the manuscript that addresses the points raised during the review process.

ACADEMIC EDITOR:Reviewer 1 focused more on the content of the study while Reviewer 2 focused on the methods. I recommend you pay attention to the comments of both reviewers as they may help you improve the manuscript, and, in particular, be sure to address the methodological concerns raised by Reviewer 2. I have no further comments.

We look forward to receiving your revised manuscript.

Kind regards,

Alberto Molina Pérez, Ph.D.

Academic Editor

PLOS ONE

Journal Requirements:

Reviewers' comments:

Reviewer's Responses to Questions

**Comments to the Author**

1. Is the manuscript technically sound, and do the data support the conclusions?

Reviewer #1: Yes

Reviewer #2: Partly

2. Has the statistical analysis been performed appropriately and rigorously? 

Reviewer #1: N/A

Reviewer #2: N/A

3. Have the authors made all data underlying the findings in their manuscript fully available?

Reviewer #1: Yes

Reviewer #2: Yes

4. Is the manuscript presented in an intelligible fashion and written in standard English?

Reviewer #1: Yes

Reviewer #2: Yes

5. Review Comments to the Author

Reviewer #1: The authors present an interview study which documents the perspectives of members of funding agencies on, amongst other elements, incentives for data sharing. The manuscript is well-written and highlights most aspects influencing data sharing. I do have some suggestions to improve the paper:

Data availability:

- "All anonymized transcript data, except for one, are available..." -> What happened with the missing data?

Introduction:

- The paper discusses a system in which scientists are not only valued by the grants they receive and the (high-impact) publications they produce, but also by other factors, such as the sharing of data. These goals are in line with https://scienceintransition.nl/en, so this initiative could be mentioned, next to the San Francisco Declaration on Research Assessment (DORA) mentioned on line 182. This OECD recommendation: https://www.oecd.org/science/inno/recommendation-access-to-research-data-from-public-funding.htm might also be interesting to mention here, as well as the Open Science registry: https://openscienceregistry.org/

- Line 61: cite the FAIR paper (https://www.nature.com/articles/sdata201618)

- I think one of the main barriers for sharing data is privacy laws such as the GDPR. In my experience, researchers were much more inclined to share data in pre-GDPR times. Different institutes (or even different people) have different interpretations of the GDPR, which makes people uncertain about whether they are allowed to share data with external parties (and in which context). This should be mentioned in the introduction or discussion section.

- What about data ownership? Aren't the patients the data owners?

Results:

- Related to the subjectivity discussion: doesn't the 'narrative CV' also have a risk? In the sense that people who are good at storytelling, using the right buzzwords, etc. might be more appreciated than researchers who perhaps published in top journals but are less able to 'sell' their story?

Discussion:

- Do you think federated/distributed learning will help with sharing data? In this way, universities/hospitals do not need to send their data to external parties, because the algorithm travels to the data instead of the other way around. See also https://pubmed.ncbi.nlm.nih.gov/32349396/ and https://www.health-ri.nl/initiatives/personal-health-train

- Data citation metrics are useful but is it not possible to misuse these just as other quantitative indicators, as you have pointed out in line 414?

Reviewer #2: The manuscript is interesting and, in general, correct. But it has some deficiencies that could improve it if they were modified.

The introduction / approach is insufficient. It is well argued why it is interesting to obtain the opinion/position of funding agencies, but it is not clear why they focus on one of the incentives ("recognition systems"; "attribution and evaluation"), so that there is no clear research question: what is problematic and how is it addressed. Exposing the results of the research in several articles can provide greater "academic" use of the research carried out, but if it is not done consistently, it can make the exposed results lose coherence and relevance. On the other hand, the review of the literature on the subject is somewhat scarce, which makes the text more of a report of research results than a scientific article.

The analysis is very descriptive and, it can be said, little analytical: some interviewees have this opinion, others have another... For example: "Some members of funding agencies voiced their frustrations that cultural discrepancies exist between the funding agencies and some research communities " (Ln. 329-332). Thus, no typology of funding agencies is made based on their position regarding the consideration or assessment of the provision of research data to the scientific community. Nor are the opinions described are related to the characteristics of the financing agencies that were considered for the sample.

In the content analysis it says that NVIVO 12 was used and that "iterative coding steps were performed by TD". It is not clear what TD is.

On the other hand, it is said that the codes emerged from the analysis ("Interview transcripts were analyze through inductive content analysis, in which codes emerge from data rather than being predetermined" Ln 155-157), but it is not clear to what extent this is so, because the interview guide is not explicit. It seems that at least the three blocks in which the results are described are interview question blocks. In this sense, it is relevant to distinguish between what was pre-established in the script and what are issues that arise in the interviews, so it would be convenient to specify it in the presentation of the analysis carried out and the results obtained.

The transcripts are conveniently anonymized, but the characteristics that define them within the sampling criteria are not indicated in the heading, the profiles of the interviewees are not specified. Although Table 1 shows the characteristics of the interviewees according to the sampling criteria, these characteristics should also be included at the beginning of the transcripts to facilitate their characterization.

Lastly, some issues that appear in the discussion should be included in the introduction/approach, because they explain the characteristics of the recognition systems under analysis (academic recognition of data sharing) and are issues that have been taken into account in the time to establish the script of the interview (interview guide). For example, ln. 360- 376.

6. PLOS authors have the option to publish the peer review history of their article (what does this mean?). If published, this will include your full peer review and any attached files.

Reviewer #1: **Yes: **Tim Hulsen

Reviewer #2: No

---

## [Author Response · Author response to Decision Letter 0]

25 Jan 2023

A: Many thanks to the reviewers for their detailed comments. We have taken note of the reviewers’ comments to substantially revise the manuscript. The introduction and discussion sections have been rewritten and expanded.

The introduction now explains better:

- Background (e.g., platforms, contributorship, indicators…). Most of this information was moved from the discussion section. We agree with reviewer 2 that this facilitates understanding the results.

- Concepts that we need to understand incentives for data sharing (e.g., attribution and evaluation systems, assumptions of the term “incentive” et cetera).

- Rationale for setting up this study

The discussion section was reworked to include following parts:

- We illustrate in more depth the state of the literature on indicators, contributorship and narrative CVs.

The methodology section now refers to Supplementary Data of the first paper we published based on this interview study about science policy interventions/measures (which contrary to incentives, we define as limiting researchers’ autonomy). 

The anonymized data in the SODHA database now gives more information on the sampling. Each anonymized transcript is tagged with funding agency characteristics.

Reviewer #1: The authors present an interview study which documents the perspectives of members of funding agencies on, amongst other elements, incentives for data sharing. The manuscript is well-written and highlights most aspects influencing data sharing. I do have some suggestions to improve the paper:

Data availability:

- "All anonymized transcript data, except for one, are available..." -> What happened with the missing data?

A: We did not upload one transcript because it could not be anonymized at all. This was a funding agency that had a very particular funding profile in terms of types of activities sponsored, which was repeated in nearly all answers to questions and was necessary to understand their perspectives. For this reason, we came to conclusion that we could not anonymize it without destroying the contents of the transcript.

Introduction:

- The paper discusses a system in which scientists are not only valued by the grants they receive and the (high-impact) publications they produce, but also by other factors, such as the sharing of data. These goals are in line with https://scienceintransition.nl/en, so this initiative could be mentioned, next to the San Francisco Declaration on Research Assessment (DORA) mentioned on line 182. This OECD recommendation: https://www.oecd.org/science/inno/recommendation-access-to-research-data-from-public-funding.htm might also be interesting to mention here, as well as the Open Science registry: https://openscienceregistry.org/

- Line 61: cite the FAIR paper (https://www.nature.com/articles/sdata201618)

- I think one of the main barriers for sharing data is privacy laws such as the GDPR. In my experience, researchers were much more inclined to share data in pre-GDPR times. Different institutes (or even different people) have different interpretations of the GDPR, which makes people uncertain about whether they are allowed to share data with external parties (and in which context). This should be mentioned in the introduction or discussion section.

A: We understand that the GDPR can be a substantial barrier for sharing. We want to mention that the effects of GDPR can in some ways be related to incentives, particularly financial ones. For instance, let’s take that GDPR makes sharing more burdensome, yet some barriers can be reasonably overcome by using privacy-preserving technologies (or other technical solutions). In this case, creating incentives for institutions to engage in data sharing might encourage them to invest into data management units. In turn, these could facilitate researchers to engage with these technologies to overcome their problems. We have decided not to include this in the manuscript. 

- What about data ownership? Aren't the patients the data owners?

A: Data ownership (in the property sense) does not really apply here. However, cohort holders are often data custodians (and data controllers according to GDPR), meaning that they are responsible for dealing appropriately with the participants data. The study participants themselves have some fundamental rights under GDPR (e.g., right to erasure, correction). We want to point out that, if consent is not the legal basis, the idea that study participants need to approve every incoming data access request does not apply. We have not discussed this further in the manuscript.

Results:

- Related to the subjectivity discussion: doesn't the 'narrative CV' also have a risk? In the sense that people who are good at storytelling, using the right buzzwords, etc. might be more appreciated than researchers who perhaps published in top journals but are less able to 'sell' their story?

A: Narrative CVs are very likely to be beneficial for persons that are good at storytelling (just like this same skill advantages researchers in grant writing). This cannot be avoided. The only reasonable thing that could prevent this is that claims can be checked somewhere (e.g., on data sharing practices or outputs). We have briefly addressed this in the discussion.

Discussion:

- Do you think federated/distributed learning will help with sharing data? In this way, universities/hospitals do not need to send their data to external parties, because the algorithm travels to the data instead of the other way around. See also https://pubmed.ncbi.nlm.nih.gov/32349396/ and https://www.health-ri.nl/initiatives/personal-health-train

A: Although these new techniques have been emerging as a potential solution to data sharing problems, we believe there are several barriers to the adoption of these techniques. Nevertheless, this discussion goes beyond the topic that is covered in the manuscript.

- Data citation metrics are useful but is it not possible to misuse these just as other quantitative indicators, as you have pointed out in line 414?

A: How they may be used depends a lot on how they are formulated (absolute, relative – process, output) and what their purposes (individual or team evaluation) are. We have tried to highlight this in the last paragraph of the discussion.

Reviewer #2: The manuscript is interesting and, in general, correct. But it has some deficiencies that could improve it if they were modified.

The introduction / approach is insufficient. It is well argued why it is interesting to obtain the opinion/position of funding agencies, but it is not clear why they focus on one of the incentives ("recognition systems"; "attribution and evaluation"), so that there is no clear research question: what is problematic and how is it addressed. 

A: The results of the interview study could be categorized along three main lines: incentives for data sharing, policy measures that restrict decision-making capacities, and funding models. We have substantially expanded the rationale for the study in the introduction.

We understand “incentives for data sharing” to mean “things that promote actors to take decisions in a particular way”. In particular, we focus on reputational incentives, which in reputation economies like academia also act as financial incentives. Incentives promote rather than compel decisions actors to make certain decisions. Other policy measures can imply restricting decision-making capacities (e.g., enforced mandates). We see these as distinct from “incentives” as they prohibit certain actions from being taken or force certain actions to be taken. 

The background for our studies is that data sharing upon motivated request often does not take place (https://www.nature.com/articles/d41586-022-01692-1). Many documents have attributed this to, amongst other reasons, a lack of incentives for data sharing. In other words, there are few reasons in terms of funding acquisition/promotion why you should share data. Therefore, they must be created. The research question is then: What may these incentives look like for cohort studies? We conceptualized this in our earlier work (e.g., contributorship models, indicators). This is why we are interested in funding agencies’ views: We wish to see what is acceptable for them to implement into their processes, such as grant evaluation.

Reviewer: Exposing the results of the research in several articles can provide greater "academic" use of the research carried out, but if it is not done consistently, it can make the exposed results lose coherence and relevance. On the other hand, the review of the literature on the subject is somewhat scarce, which makes the text more of a report of research results than a scientific article.

A: We have separated the three topics (incentives, policy measures, funding models) for the sake of understandability. An earlier version of the manuscript of all three topics together was about 15.000 words. After discussing it internally, we agreed that it was not comprehensible to outside researchers. In particular, the major difficulty was (and still is) making the interconnections between different “interventions” in future policy frameworks visible. When we report on each topic individually, it becomes more understandable. Furthermore, the results were very rich and splitting this up in multiple parts allowed us to explain more in-depth the results

The paper on policy measures has already been published. 

Reviewer: The analysis is very descriptive and, it can be said, little analytical: some interviewees have this opinion, others have another... For example: "Some members of funding agencies voiced their frustrations that cultural discrepancies exist between the funding agencies and some research communities " (Ln. 329-332). Thus, no typology of funding agencies is made based on their position regarding the consideration or assessment of the provision of research data to the scientific community. Nor are the opinions described are related to the characteristics of the financing agencies that were considered for the sample.

A: The positions of funding agencies were not clear cut as to allow summarizing them according to a fixed typology. The main issue is that we are discussing future orientations and that funding agencies made clear that there is no one “route” that they will take. Instead, they would state things could be done in several ways, that it was not yet clear, that they may consider X and Y, that they are discussing X, Y and Z and so on. Funding agencies statements also were often influenced by discussions around data citation (which amongst all metrics for data sharing is the most advanced in development) and data repositories (e.g., related to certification, funding models). The catch here is that data sharing platforms that we describe function quite differently from data repositories and data citation is, so far, not being integrated into workflows. We therefore provide descriptions of what interviewees said in terms of general principles of metrics (e.g., should be related directly to object, cite within narrative CV, process indicators on open science…). 

It is correct that we do not identify patterns according to the types of funding agencies. We did not see clear differences in opinions based on the characteristics in this sample.

Our view is that this descriptive approach still allows us to gain insight into what could be acceptable to funding agencies. A detailed analysis of funding agencies according to their characteristics is not necessary to do this. 

Reviewer: In the content analysis it says that NVIVO 12 was used and that "iterative coding steps were performed by TD". It is not clear what TD is.

A: We meant the first author. We have altered this line.

Reviewer: On the other hand, it is said that the codes emerged from the analysis ("Interview transcripts were analyze through inductive content analysis, in which codes emerge from data rather than being predetermined" Ln 155-157), but it is not clear to what extent this is so, because the interview guide is not explicit. It seems that at least the three blocks in which the results are described are interview question blocks. In this sense, it is relevant to distinguish between what was pre-established in the script and what are issues that arise in the interviews, so it would be convenient to specify it in the presentation of the analysis carried out and the results obtained.

A: We do not really agree with the reviewer that the blocks in which results were described (i.e., evaluation systems, alternative attribution systems, and role of research community and international alignment) match the interview questions. The interview questions are much more concrete in our view than the codes (i.e., We did not ask “How can attribution systems be altered to encourage data sharing?”). The questions in the interview guide do not explicitly describe this, but they are formulated in more practical terms: monitoring, rewarding, DMPs et cetera. We also want to point out that, as the interviews were semi-structured, the flow of the interviews did not lead us to go over each question one by one. Interviews were much more fluid, with many follow-up questions being asked about what participants bring up.

The coding process was not done explicitly with prior categories in mind. We did not try to map parts of the interview transcripts to pre-existing codes or categories. Of course, the conceptualization of categories in the coding process is inevitably influenced by pre-existing ideas (e.g., what the term “incentive” means). 

We now refer in the methodology section to the first paper on policies to stimulate data sharing, where we uploaded an extended method as Supplementary Data. 

Reviewer: The transcripts are conveniently anonymized, but the characteristics that define them within the sampling criteria are not indicated in the heading, the profiles of the interviewees are not specified. Although Table 1 shows the characteristics of the interviewees according to the sampling criteria, these characteristics should also be included at the beginning of the transcripts to facilitate their characterization.

A: We have updated the metadata of transcript files to include the profiles of the funding agencies as tags. This makes it possible to reuse them without looking at the table.

Reviewer: Lastly, some issues that appear in the discussion should be included in the introduction/approach, because they explain the characteristics of the recognition systems under analysis (academic recognition of data sharing) and are issues that have been taken into account in the time to establish the script of the interview (interview guide). For example, ln. 360- 376.

A: We have moved these lines from discussion to introduction. Furthermore, we have expanded the introduction and discussion substantially. It should now be much more understandable.

---

## [Decision Letter · Decision Letter 1]

28 Feb 2023

Reward Systems for Cohort Data Sharing: An Interview Study with Funding Agencies

PONE-D-22-23763R1

Dear Dr. Devriendt,

We’re pleased to inform you that your manuscript has been judged scientifically suitable for publication and will be formally accepted for publication once it meets all outstanding technical requirements.

Kind regards,

Elizabeth McGill

Academic Editor

PLOS ONE

**Comments to the Author**

Congratulations on your manuscript; the reviewers have two remaining minor comments. Please include the additional references as requested by Reviewer #1. Reviewer #2 has requested that you include the interview questions in the manuscript text, although this is not a mandatory addition for publication.  

Reviewer #1: (No Response)

Reviewer #2: All comments have been addressed

2. Is the manuscript technically sound, and do the data support the conclusions?

Reviewer #1: Yes

Reviewer #2: Yes

3. Has the statistical analysis been performed appropriately and rigorously? 

Reviewer #1: N/A

Reviewer #2: N/A

4. Have the authors made all data underlying the findings in their manuscript fully available?

Reviewer #1: Yes

Reviewer #2: Yes

5. Is the manuscript presented in an intelligible fashion and written in standard English?

Reviewer #1: Yes

Reviewer #2: Yes

6. Review Comments to the Author

Reviewer #1: All comments have been addressed.

Just a minor remark:

- "Some examples of platforms ... [(5,6)]  Please include references to all platforms instead of just two.

Reviewer #2: I still think that it would be interesting to specify in the text which questions were addressed in the interview guide in order to be able to more accurately assess what arose spontaneously in the interviews. However, this is a minor issue that does not affect the validity or interest of the data, so it is not cause for rejection. Congratulations on the research and the manuscripts.

7. PLOS authors have the option to publish the peer review history of their article (what does this mean?). If published, this will include your full peer review and any attached files.

Reviewer #1: **Yes: **Tim Hulsen

Reviewer #2: **Yes: **Jorge Ruiz Ruiz (IESA, CSIC)

---

## [Editor Report · Acceptance letter]

16 Mar 2023

PONE-D-22-23763R1 

Reward Systems for Cohort Data Sharing: An Interview Study with Funding Agencies 

Dear Dr. Devriendt:

I'm pleased to inform you that your manuscript has been deemed suitable for publication in PLOS ONE. Congratulations! Your manuscript is now with our production department. 

Kind regards, 

on behalf of

Dr. Elizabeth McGill 

Academic Editor

PLOS ONE